# Effect of Sweet Corn Residue on Micronutrient Fortification in Baked Cakes

**DOI:** 10.3390/foods8070260

**Published:** 2019-07-16

**Authors:** Yu-Xia Lao, Yu-Ying Yu, Gao-Ke Li, Shao-Yun Chen, Wu Li, Xu-Pu Xing, Xue-Min Wang, Jian-Guang Hu, Xin-Bo Guo

**Affiliations:** 1School of Food Science and Engineering, South China University of Technology, Guangzhou 510640, China; 2Key Laboratory of Crops Genetics Improvement of Guangdong Province, Crop Research Institute, Guangdong Academy of Agricultural Sciences, Guangzhou 510640, China

**Keywords:** sweet corn, cake, vitamin E, carotenoids, folate

## Abstract

Owing to the concept of modern life and health, traditional baked foods are seeking transition. In this study, sweet corn residue (SCR) was used to replace wheat flour in cakes. We conducted sensory evaluation and texture analysis to assess sample quality. Also, we simulated digestion in vitro, and determined the content of total sugar and dietary fiber. The content of vitamin E and carotenoids were determined by High Performance Liquid Chromatography (HPLC), and the content of folate was determined by a microbiological method. With the increase of SCR, the content of dietary fiber, folate, vitamin E, and carotenoids significantly increased, and the digestive characteristics improved simultaneously. Based on the above evaluations, SCRC2 (sweet corn residue cake with 60% substitution) had similar sensory quality to the control (pure wheat flour cake) and had the characteristics of slow digestibility and high micronutrients.

## 1. Introduction

Traditional baked foods, being rich in fat and sugar, are not in line with modern people’s pursuit of health. Besides, excessive energy intake makes people prone to chronic diseases such as obesity [1]. Studies have illustrated that increased crude food consumption is associated with reduced risk of cardiovascular disease, type 2 diabetes, and some cancers [2,3]. Meanwhile, whole grains provide various healthy phytochemicals (including phenolics, carotenoids, and vitamin E) and dietary fiber.

People improved the whole grain content of baked food by adding different ways to obtain healthier delicacies. Gómez et al. [4] made cakes with wheat, rye, barley, and other whole wheat flours to reduce the risk of diabetes. Žilić et al. [5] added anthocyanin-rich corn flour to cookies to improve their antioxidant activity. Omoba et al. [6] developed whole grain sorghum-soya and pearl millet-soya composite biscuits as dietary fiber supplements. Others increased daily intake of lutein and whole grain foods by processing baked food containing high lutein and lutein-fortified whole wheat flour [7].

Sweet corn is a common cereal, with access to phytochemicals such as dietary fiber, vitamin E, carotenoids, and folate. Sweet corn residue (SCR), a byproduct of sweet corn juice extrusion, has similar characteristics. The dietary fiber in SCR is neither digested nor absorbed, and adequate intake contributes to reduce blood sugar and prevent colorectal cancer [8]. Vitamin E is a lipid-soluble vitamin, which is conducive to reducing cardiovascular risk, immune regulation, and is anti-allergy [9]. Carotenoids have good biological activity. Adequate intake of carotenoids can reduce the risk of breast cancer, and cardiovascular and eye diseases [10,11]. Folate is essential for human metabolism. Low folate can lead to various adverse reactions and diseases, for instance, inadequate folate intake in early pregnancy can easily cause fetal neural tube defects and malformation [12,13].

In our study, SCR was used to replace wheat flour to make cakes. The sensory evaluation, texture characteristics, and digestive characteristics of sweet corn residue cake (SCRC) with different additions were compared. The composition and content of dietary fiber, vitamin E, carotenoids, and folate in cakes were determined. The purpose of our study was to increase the added value of SCR and develop a cake with slow digestibility and high nutrition for consumers concurrently.

## 2. Materials and Methods

### 2.1. SCR Preparation

The variety YT28 of fresh sweet corn kernels was obtained from the Crop Science Institute of Guangdong Academy of Agricultural Sciences (Guangzhou, China). We dried the residue obtained from the squeezing of sweet corn kernels, crushed it into powder by a pulverizer (Yongkang Zhaoshen Electrical Appliances Co., Ltd., Yongkang, China), then sifted it through 40 meshes. The SCR was packed and sealed for using.

### 2.2. Cake Formulation and Preparation

SCR was prepared in our laboratory, meanwhile, other raw materials were purchased from local supermarkets. Corresponding improvements have been made to the method of Xian et al. [14], and we made batter in the light of the formula in Table 1 and baked it. The baked cakes were cooled and sealed for further inspection.

All the samples were numbered according to composition. The pure wheat flour was the control, and SCR with 30%, 60%, 90%, and 100% substitution was named SCRC1, SCRC2, SCRC3, and SCRC4.

### 2.3. Sensory Evaluation

Modified in ISO 8587:2006 [15] and ISO 5495:2005 [16], sensory evaluation was organized. We established the evaluation criteria of color, texture, shape, taste, and flavor by the predicted results. The full score of each index was 20 points; a higher score represented closer to the standard. In the sensory evaluation laboratory, we presented 5 numbered cake samples to 12 trained tasters. They completed the evaluation independently and recorded results. Data were reported as mean ± SD (*n* = 12).

### 2.4. Texture Profile Analysis of Cakes

Modifying Brites et al. [17], texture characteristics of cakes were measured with a texture analyzer (TA.XTplusC Texture Analyser, Stable Micro Systems, London, UK), and cubes of 20 mm cake cores were used as the test samples. Texture profile analysis (TPA) was performed at a pretest speed of 2 mm·s^−1^, test speed of 1 mm·s^−1^, time interval of 5 s, and strain deformation of 50% by an aluminum probe P/25R. Hardness, resilience, cohesion, springiness, gumminess, and chewiness were calculated from the TPA graphic. Data were reported as mean ± SD (*n* = 3).

### 2.5. Determination of Total Sugar

The total sugar content of the samples was determined by the anthrone-sulfate method [18]. Anhydrous glucose was used as standard material and drew a standard curve. We crushed and dissolved samples and took the supernatant after centrifugation. The reaction liquid was obtained by a reaction of diluted supernatant with anthrone. The absorbance value was determined by visible-ultraviolet spectrophotometer at 620 nm. The total sugar content was expressed in mg/100 g sample (mean ± SD, *n* = 3).

### 2.6. Determination of Dietary Fiber

Modifying Kamotho et al. [19], dietary fiber was determined by an SLQ-6 semi-automatic fiber analyzer (Shanghai Fiber Inspection Instrument Co., Ltd., Shanghai, China). We added samples, then preheated the acid-based solution and ultrapure water. We stopped the reaction after digesting for 35 min and washed the residue. We dried and weighed the remaining solids and then calculated the content. 

### 2.7. In Vitro Digestion

Modifying Gao et al. [20,21], we made corresponding improvements. Referring to white bread (Blank), we used pepsinase, alpha-amylase, and buffer to simulate digestion in vitro. The digestive samples were taken at 0, 15, 60, 90, 120, 150, and 180 min respectively. We deactivated the enzymes and then diluted them. The content of reducing sugar was determined by 3,5-Dinitrosalicylic acid (DNS), and the absorbance at 540 nm was measured by the enzyme label. The results were expressed as mean ± SD (*n* = 3).

### 2.8. Extraction and Determination of Vitamin E

Vitamin E in cake samples was extracted and determined referring to Xie et al. [22,23]. For the saponified cake samples, we extracted a mixture with n-hexane/ethyl acetate (9:1 *v/v*) and then collected an organic layer. After evaporation, the residues were dissolved in isopropanol (1%) n-hexane solution for HPLC. Equipped with a Waters 2475Multi λ Fluorescence Detector and Waters 515 HPLC pump, an NP-HPLC system was chosen for vitamin E determination. Additionally, an Agilent ZORBAX RX-SIL column was used. By the comparison of retention time between samples and standards, the identification of vitamin E was carried out. The results were reported as µg/100 g sample (mean ± SD, *n* = 3).

### 2.9. Extraction and Determination of Carotenoids

Consulting Liu et al. [24], the extraction and determination of carotenoids in SCRC was performed. In this experiment, the extraction of carotenoids was the same as vitamin E. Separation and quantification were performed by a HPLC system (Waters Corporation, Milford, MA, USA), with YMCTM carotenoid 30, 4.5 × 250 mm column, and a photodiode array detector. Comparing the storage time of standard carotenoid samples, the content of carotenoid was determined. The results were reported as µg/100 g sample (mean ± SD, *n* = 3).

### 2.10. Extraction and Determination of Folate

Modifying Shan et al. [23,25], the extraction and determination of folate in cakes was conducted. We crushed samples to extract folate, then determined the content. Briefly, the extracts and Difco™ Folic Acid Assay Medium (Sparks, MD, USA) was mixed. Additionally, for inoculation, an *Enterococcus hirae* (ATCC^®^ 8043™, American Type Culture Collection, Manassas, VA, USA) suspension was used. The content of folate was calculated according to folic acid standard curves (Sigma, St. Louis, MO, USA) after absorbance values were measured at 660 nm. The results of folate content were shown as µg/100 g (mean ± SD, *n* = 3).

### 2.11. Statistical Analysis

In this study, statistical analysis was conducted by the Origin Lab Corporation (Northampton, MA, USA). With SPSS software 21.0 (SPSS, Chicago, IL, USA), we analyzed the differences among samples by Tukey’s multiple comparison test (*p* < 0.05), the correlation between SCR addition, and evaluation results by Pearson coefficient as well.

## 3. Results

### 3.1. Effect of SCR on Sensory Quality of Cake

Table 2 reported the results of evaluation of color, texture, shape, taste, and flavor. In Table 2, we observed that there was no significant difference among SCRC1, SCRC2, and the control. The texture and shape of SCRC1 and SCRC2 were slightly inferior to that of the control. However, they had more uniform color and a rich sweet corn flavor. SCRC3 was similar to SCRC4, but abundant SCR caused a sag in the shape and a dark color. Consequently, their scores decreased significantly. Generally speaking, when the SCR content was less than 60%, the negative effect of its substitution on cake quality could be neglected. Instead, consumers preferred the natural corn fragrance it brought.

### 3.2. Effect of SCR on Texture Characteristics of Cake

The test results with the texture analyzer (TA.XTplusC Texture Analyser, Stable Micro Systems, London, UK)are shown in Table 3. Observation of Table 3 showed that when the SCR content was less than 60%, the hardness and chewiness of the cake decreased, and the gumminess became poor, but the springiness and cohesion changed slightly, or even improved. When the SCR content reached 90% or 100%, the cake became harder and more difficult to chew. Its springiness and resilience decreased significantly, and the overall quality of the cake deteriorated.

### 3.3. Dietary Fiber in Cakes

The results were 3.67% and 1.75%, respectively. In our study, the baking temperature was far below the requirement of dietary fiber decomposition. Thus, we speculated that the loss of dietary fiber in SCR during the cake making process could be negligible, and the fiber content in cake products could be converted by raw materials. Converted, the content of dietary fiber in the control, SCRC1, SCRC2, SCRC3, and SCRC4 were 324.6, 423.5, 528.3, 621.,7 and 656.1 mg/100 g, respectively.

### 3.4. In Vitro Digestion of Cakes

Figure 1 reflected the amount of reducing sugar released in 180 min, which was obtained from starch hydrolyzed by digestive enzymes. From Figure 1, the reducing sugar of all samples was released rapidly in the first 20 min, but the content of reducing sugar released during subsequent digestion was strikingly different. When digestion in vitro stopped, the AUC (area under the curve, which meant total release of reduced sugar during 180 min of in vitro digestion) of the control, SCRC1, SCRC2, SCRC3, and SCRC4 only accounted for 80%, 56%, 41%, 8.7%, and 6.9% of Blank. Thus, we inferred that the Glycemic Index (GI) value of cake samples decreased significantly with the increase of SCR. Simultaneously, the content of soluble reducing sugar in the control, SCRC1, SCRC2, SRC3, SCRC4, and Blank were 69.9, 64.3, 48.5, 40.4, 29.9, and 105.6 mg/g, respectively. One gram of soluble reducing sugar provided about 16 kJ of calories. It could be inferred that the calories provided by reducing sugar in 100 g samples were 1118.4, 1028.8, 776.0, 646.4, 478.4, and 1689.6 kJ, sequentially. However, the total sugar content determined by the anthrone method was inconsistent with the results of in vitro digestion, as shown in Table 4.

### 3.5. Composition and Content of Vitamin E in Cakes

The composition and content of vitamin E detected in cakes are indicated in Table 5. α-, β-, γ-, δ-tocopherols and α-, β-, γ-, δ-tocotrienols were detected in this study. The content of γ-tocopherol was the highest—about half of the total—followed by α-tocopherol, and δ-tocotrienol was the lowest. The cakes in our study all contained high vitamin E content owing to the use of corn oil. The result indicated that the vitamin E content of SCRC was about 1.5 times that of the control. In other words, the addition of SCR could significantly increase the vitamin E content of cake.

### 3.6. Composition and Content of Carotenoids in Cakes

The composition and content of carotenoids detected in cakes are shown in Table 6. Only lutein, zeaxanthin, α-, β-cryptoxanthin, and β-carotene were detected in the control, and the total content was low. Lutein, zeaxanthin, α-, β-cryptoxanthin, and α-, β-, ε-, (6R)-δ-carotene were all detected in SCRC. Lutein, zeaxanthin, and α-cryptoxanthin were the three most abundant carotenoids detected, which were consistent with the composition of sweet corn carotenoids. Comparing the results, we observed that the highest carotenoid content of SCRC was 3.5 times that of the control. The carotenoid content in cake was positively correlated with SCR content.

### 3.7. Folate Content in Cakes

The content of folate detected in cakes is shown in Figure 2. As shown, the folate content of SCRC increased significantly with the growth of SCR. The folate content of the SCRC4 reached 776.6 μg/100 g, which was 1.7 times that of the control. 

### 3.8. Correlation between SCR Addition and Detection Indicators

As shown in Table 7, the addition of SCR was negatively correlated with sensory scores, resilience and cohesion, but had no significant effect on other texture indexes. In addition, the correlation coefficient between dietary fiber content and SCR addition was 1, which indicated that the increase of dietary fiber in cakes was mainly due to SCR. The GI value of the cake was significantly reduced by dietary fiber brought in by SCR.

Through Table 8, there was a positive correlation between SCR addition and vitamin E content. The more SCR that was added, the more carotenoids were in cakes, and the majority of its components also showed this pattern. Folate content in cakes increased linearly with the increase of SCR.

## 4. Discussion

### 4.1. The Relationship between Sensory and Textural Characteristics

Sensory evaluation of cake quality is based on individual judgment of evaluation criteria, which is subjective. However, it can predict consumers’ acceptance and preference to an extent. Besides, the texture of cake can be predicted by the amount of eggs, milk, grease, flour, and other ingredients, and then it depends on the processing conditions, such as batter mixing and baking temperature and methods.

Singh et al. [26] indicated that cake with corn bran had a low taste score, but the substitution level of 10% and 20% had no significant effect on the overall acceptability score of the cake. In our study, we found that the cake became harder and stickier with springiness and resilience decreasing significantly when the SCR content reached 90% or 100%. Correspondingly, the scores of taste and texture in sensory evaluation decreased. Also, the sensory and texture characteristics of SCR1 and SCR2 were similar to those of the control with the same scores. Our results showed that SCR could be a suitable substitute for wheat flour without significantly affecting the sensory properties and texture characteristics of cakes, such as SCRC1 and SCRC2.

### 4.2. Interaction between Dietary Fiber and Digestion In Vitro

Researchers have confirmed that the nutritional value of cakes improved by incorporating dietary fiber properly [27]. In our study, when the substitution amount of SCR reached 60%, the dietary fiber content of SCR2 was 1.6 times that of the control, and it maintained a good taste. Additionally, dietary fiber was used to further reduce GI content in starchy foods, such as pasta [28]. Our results showed that the AUC decreased while dietary fiber increased. The growth of dietary fiber in samples mainly came from the increase of SCR, which confirmed that the addition of SCR reduced GI values. The AUC of SCRC2 was only 41% of Blank, which indicated that it was a delicious cake with slow digestibility.

However, there was no significant correlations among the measured total sugar content, dietary fiber, and AUC. The main reasons were as follows. Firstly, the total sugar determined by the anthrone method contained soluble reducing sugar and non-reducing sugar, such as dietary fiber. Our samples were rich in dietary fiber, which accounted for a large part of the total sugar content. Secondly, some studies pointed out that the combination of dietary fiber in pasta formulations showed a significant reducing sugar decrease in vitro [29]. In other words, SCR brought in abundant dietary fibers, which could antagonize the release process of reducing sugar during in vitro digestion. 

Based on the results and discussion above, we rated that SCR2 was a viable and delicious cake formula with rich dietary fiber and a lower GI value, owing to the substitution of SCR.

### 4.3. Micronutrient Fortification Effects in Cakes 

Many surveys revealed that people usually intake vitamin E with antioxidant activity through vegetables and fruits [9,30]. Moreover, studies have found that proper processing does not significantly affect vitamin E content and activity [31]. Our detected results agreed with it, which showed that all the cakes in this study contained a high content of vitamin E. The vitamin E in cakes mainly came from corn oil and SCR. From the results, we knew that the content of vitamin E increased with the addition of SCR under the premise of the same use of corn oil. The content of vitamin E in SCRC2 was 1.4 times that of the control, which meant it enhanced vitamin E to some extent by adding SCR. 

Carotenoids are common substances in the food and health products industry. They have the characteristics of anti-oxidation, immune regulation, anti-cancer, and anti-aging [32]. Some studies have indicated that people can obtain high-content carotene by making whole wheat or corn baked food [33]. Similarly, we found that the content of carotenoids increased significantly with the increase of SCR. According to the results, there were four carotenoid components in the control but seven in SCRC. It indicated that the addition of SCR enriched the composition of carotenoids in cakes while improving content. Compared with the control, the increase of carotenoids was 170% in SCRC2. In other words, SCRC2 achieved high-efficiency enhancement of carotenoids and had potential as a supplement. Consumers would prefer to supplement natural carotenoids with delicious SCRC2 rather than synthetic tablets.

Researchers have pointed out that folate had a small amount of loss in the process of baking and storage [34,35]. Similarly, SCR lost parts of folate during cake baking. However, according to our results, adding SCR was still an economical and effective way to fortify folate. Folate can prevent fetal neural tube defects and is one of the indispensable nutrients for fetal growth and development [36]. The recommended daily supplement of folate for pregnant women is 400 μg per day. In our study, 100 g of SCRC2 contained 692 μg folate, which could meet this requirement within the normal consumption range. Besides, too much folic acid is a burden on the human liver, so experts suggest that biofortification should be used to increase folate content in staple foods or fruits and vegetables as a safer way to supplement folate [36]. The folate content in cakes fortified by SCR coincided with this suggestion.

## 5. Conclusions

In our study, different proportions of SCR were used instead of wheat flour to make cakes. We found that SCR could be used as an excellent carrier to fortify dietary fiber, carotenoids, vitamin E, and folate for baking foods. Through overall evaluation, SCRC2 was the best formulation with excellent food quality. Also, it had the characteristics of slow digestibility and high micronutrient content, which would be popular with consumers.

## Figures and Tables

**Figure 1 foods-08-00260-f001:**
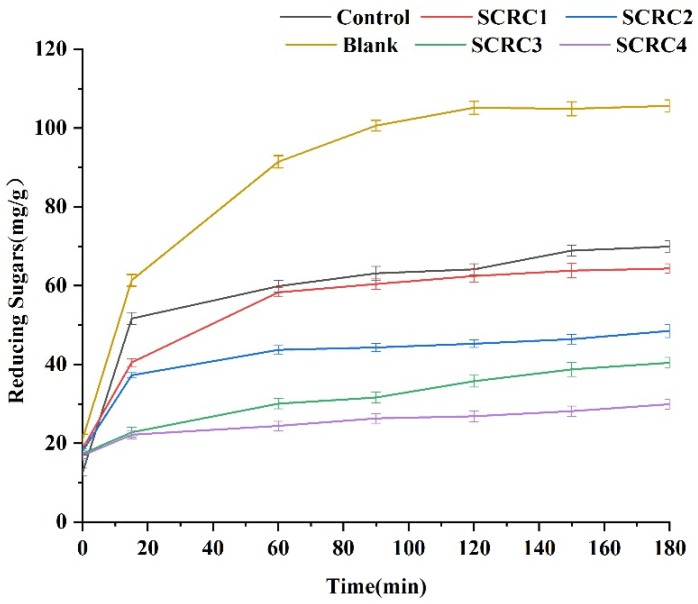
Amount of reducing sugars released from Sweet corn residue (SCR) cakes during in vitro digestion.

**Figure 2 foods-08-00260-f002:**
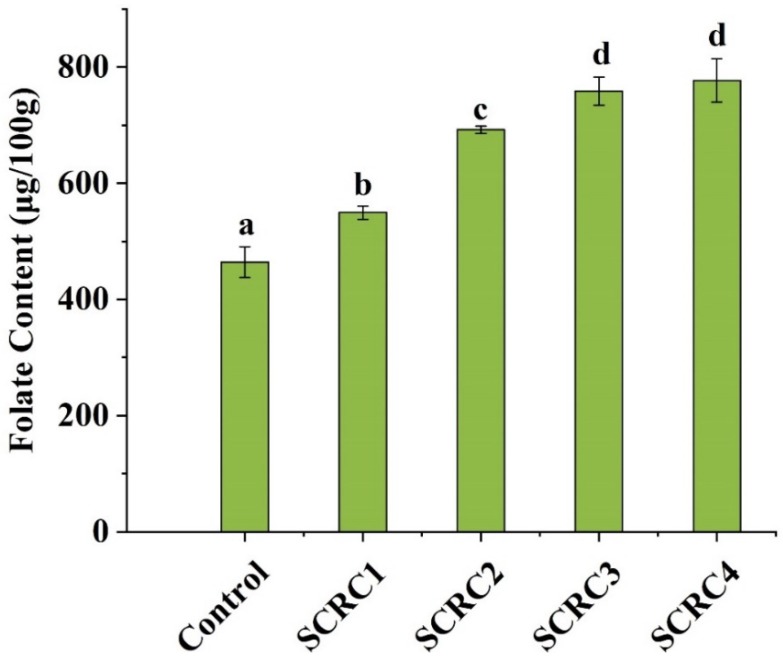
Content of folate in cakes with SRC addition. Bars with different letters in each fraction are significantly different (*p* < 0.05).

**Table 1 foods-08-00260-t001:** Formulation of sweet corn residue (SCR) cakes.

Ingredients (g)	Control	SCRC1	SCRC2	SCRC3	SCRC4
SCR	0	15	30	45	50
Low-gluten flour	40	28	16	4	0
High-gluten flour	10	7	4	1	0
Egg	95	95	95	95	95
Sugar	36	36	36	36	36
Salt	0.3	0.3	0.3	0.3	0.3
Soybean lecithin	0.5	0.5	0.5	0.5	0.5
Baking powder	2.5	2.5	2.5	2.5	2.5
Corn oil	25	25	25	25	25
Milk	60	65	65	70	70

**Table 2 foods-08-00260-t002:** Effect of SCR addition on the sensory quality of cakes. The five indicators (color, texture, shape, taste, flavor) have the same proportions.

Products	Color	Texture	Shape	Taste	Flavor	Total Score
Control	14.18 ± 2.36	17.55 ± 2.11	16.64 ± 1.96	18.00 ± 1.61	14.64 ± 3.53	81.00 ± 7.81
SCRC1	15.82 ± 1.72	16.00 ± 2.00	16.00 ± 2.28	16.00 ± 3.29	16.27 ±2.65	80.09 ± 8.61
SCRC2	15.36 ± 2.94	15.73 ± 2.69	15.82 ± 2.56	15.36 ± 2.94	17.18 ± 2.14	79.45 ± 10.93
SCRC3	14.55 ± 3.50	14.09 ± 2.91	13.82 ± 3.34	12.55 ± 4.27	15.27 ± 2.971	70.27 ± 14.46
SCRC4	14.45 ± 5.16	13.82 ± 3.37	14.09 ± 4.30	12.09 ± 3.51	15.00 ± 3.46	69.45 ± 15.45

**Table 3 foods-08-00260-t003:** Effect of Sweet corn residue (SCR) addition on the texture characteristics of cakes.

Products	Hardness (g)	Resilience (%)	Cohesion	Springiness (%)	Gumminess (g)	Chewiness (g)
Control	269.6 ± 14.2b	28.00 ± 1.17c	0.7245 ± 0.016bc	78.86 ± 3.76ab	195.8 ± 8.3bc	153.0 ± 6.6bc
SCRC1	195.1 ± 7.1a	30.93 ± 0.45d	0.7497 ± 0.0178c	84.76 ± 6.70b	144.8 ± 5.3a	123.3 ± 11.5a
SCRC2	259.9 ± 12.2b	27.50 ± 0.93c	0.7122 ± 0.0154b	75.41 ± 1.98a	183.4 ± 8.0b	135.5 ± 10.4ab
SCRC3	310.8 ± 26.1c	24.54 ± 0.44b	0.6741 ± 0.0053a	76.08 ± 0.40a	201.0 ± 16.7c	159.6 ± 11.9c
SCRC4	388.8 ± 20.0d	22.01 ± 0.80a	0.6598 ± 0.0083a	73.81 ± 0.52a	256.9 ± 9.4d	188.2 ± 9.0d

Means in the same column with different letters are significantly different (*p* < 0.05).

**Table 4 foods-08-00260-t004:** Effect of SCR on total sugar and in vitro digestion profile in samples.

Products	Total Sugar (mg/g)	AUC (Area under the Curve)
Control	158.9 ± 9.1c	4133 ± 54e
SCRC1	169.2 ± 1.9d	2895 ± 50d
SCRC2	135.6 ± 3.6b	2115 ± 10c
SCRC3	165.0 ± 5.5cd	454.9 ± 16.5b
SCRC4	168.2 ± 1.9d	359.5 ± 17.5a
Blank	112.5 ± 0.4a	5212 ± 35f

Means in the same column with different letters are significantly different (*p* < 0.05).

**Table 5 foods-08-00260-t005:** Composition and content (μg/100 g) of vitamin E in cakes.

Compositions	Control	SCRC1	SCRC2	SCRC3	SCRC4
α-T	615.4 ± 13.8a	1022 ± 1b	937.0 ± 33.7b	946.4 ± 53.9b	989.1 ± 52.3b
α-T3	73.34 ± 3.61a	146.2 ± 0.2b	193.3 ± 3.4c	221.0 ± 10.5d	232.0 ± 12.4d
β-T	39.56 ± 0.04a	48.24 ± 0.63b	42.21 ± 2.30a	39.31 ± 1.91a	42.12 ± 1.57a
γ-T	1056 ± 12a	1367 ± 1d	1142 ± 26b	1168 ± 36b	1251 ± 39c
γ-T3	127.0 ± 8.5a	245.8 ± 1.1b	318.4 ± 0.9c	381.8 ± 16.6d	393.8 ± 17.9d
δ-T	47.27 ± 0.84c	51.05 ± 0.62d	39.96 ± 0.35a	42.02 ± 1.34b	48.67 ± 1.49c
δ-T3	14.66 ± 1.02a	17.29 ± 0.48b	18.48 ± 0.59bc	20.44 ± 1.19d	19.28 ± 0.23cd
Total	1973 ± 41a	2878 ± 2cd	2692 ± 63b	2820 ± 115bc	2976 ± 119d

Means in the same column with different letters are significantly different (*p* < 0.05); α-T, α-T3, β-T, γ-T, γ-T3, δ-T, δ-T3 mean α-tocopherol, α-tocotrienol, β-tocopherol, γ-tocopherol, γ-tocotrienol, δ-tocopherol, and δ-tocotrienol, sequentially.

**Table 6 foods-08-00260-t006:** Composition and content of carotenoids in cakes.

Compositions	Control	SCRC1	SCRC2	SCRC3	SCRC4
Lut	12.68 ± 0.55a	31.37 ± 1.29b	51.54 ± 2.64c	69.31 ± 4.04e	64.32 ± 2.73c
Zea	14.05 ± 0.77a	20.14 ± 0.24b	26.14 ± 1.03c	31.73 ± 1.81d	31.16 ± 3.08d
α-Cry	3.28 ± 0.10a	10.33 ± 0.05b	15.69 ± 0.28c	20.68 ± 1.02d	19.63 ± 0.56c
β-Cry	13.38 ± 1.38d	7.74 ± 0.36a	9.67 ± 0.11b	11.58 ± 0.18c	10.44 ± 0.71bc
ε-Car	ND	6.67 ± 0.09a	8.00 ± 0.24b	8.89 ± 0.37c	8.37 ± 0.18b
α-Car	ND	3.45 ± 0.23a	4.65 ± 0.24b	5.72 ± 0.67c	5.34 ± 0.34bc
β-Car	2.35 ± 0.02d	3.14 ± 0.02a	3.95 ± 0.20b	4.91 ± 0.20c	4.59 ± 0.10bc
(6R)-δ-Car	ND	3.66 ± 0.01a	5.38 ± 0.05b	6.35 ± 0.15c	6.14 ± 0.23c
Total μg/100 g	45.74 ± 2.11a	86.50 ± 1.24b	125.0 ± 4.6c	159.2 ± 8.1e	145.0 ± 7.9d

Means in the same column with different letters are significantly different (*p* < 0.05). ND means “not detected”; Lut, Zea, α-Cry, β-Cry, ε-Car, α-Car, β-Car, (6R)-δ-Car, and γ-Car mean lutein, zeaxanthin, α-cryptoxanthin, β-cryptoxanthin, ε-carotene, α-carotene, β-carotene, (6R)-δ-carotene, and γ-carotene, sequentially.

**Table 7 foods-08-00260-t007:** Correlation between SCR addition and texture, sensory evaluation and digestibility.

	Addition	Hardness	Resilience	Cohesion	Springiness	Gumminess	Chewiness	Sensory Scores	Total Sugar	Fiber	AUC
Addition	1.000										
Hardness	0.719	1.000									
Resilience	−0.807	−0.986 **	1.000								
Cohesion	−0.856	−0.950 *	0.989 **	1.000							
Springiness	−0.676	−0.847	0.871	0.867	1.000						
Gumminess	0.655	0.996 **	−0.970 **	−0.925 *	−0.845	1.000					
Chewiness	0.599	0.978 **	−0.940 *	−0.890 *	−0.741	0.984 **	1.000				
Sensory score	−0.879 *	−0.856	0.909 *	0.936 *	0.639	−0.814	−0.826	1.000			
Total sugar	0.122	0.222	−0.190	−0.172	0.309	0.207	0.368	−0.456	1.000		
Fiber	1.000 **	0.717	−0.805	−0.854	−0.681	0.652	0.594	−0.872	0.106	1.000	
AUC	−0.993 **	−0.707	0.800	0.855	0.626	−0.641	−0.602	0.911 *	−0.213	−0.991 **	1.000

** means significant correlation at 0.01 level, * means significant correlation at 0.05 level; AUC means area under the curve, which meant total release of reduced sugar during 180 min of in vitro digestion.

**Table 8 foods-08-00260-t008:** Correlation between SCR addition and detected micronutrient components.

	Addition	α-T	α-T3	β­T	γ­T	γ­T3	δ­T	δ­T3	Total VE	Lut	Zea	α-Cry	β-Cry	ε-Car	α-Car	β-Car	(6R)-δ-Car	Total Car	Folate
Addition	1.000																		
α-T	0.709	1.000																	
α-T3	0.981 **	0.813	1.000																
β­T	−0.185	0.532	−0.045	1.000															
γ­T	0.238	0.780	0.333	0.871	1.000														
γ­T3	0.987 **	0.800	0.998 **	−0.070	0.326	1.000													
δ­T	−0.316	−0.006	−0.347	0.575	0.579	−0.336	1.000												
δ­T3	0.944 *	0.794	0.968 **	−0.083	0.311	0.973 **	−0.414	1.000											
Total VE	0.782	0.988 **	0.859	0.457	0.764	0.851	0.034	0.831	1.000										
Lut	0.983 **	0.724	0.983 **	−0.197	0.206	0.989 **	−0.439	0.982 **	0.778	1.000									
Zea	0.990 **	0.748	0.990 **	−0.158	0.248	0.995 **	−0.393	0.979 **	0.804	0.998 **	1.000								
α-Cry	0.983 **	0.767	0.991 **	−0.133	0.266	0.995 **	−0.403	0.987 **	0.817	0.998 **	0.999 **	1.000							
β-Cry	−0.184	−0.814	−0.348	−0.898 *	−0.870	−0.317	−0.237	−0.313	−0.736	−0.204	−0.235	−0.266	1.000						
ε-Car	0.912	−0.629	0.936	−0.989 *	−0.797	0.940	−0.622	0.967 *	−0.056	0.988 *	0.971 *	0.981 *	0.994 **	1.000					
α-Car	0.952 *	−0.523	0.962 *	−0.962 *	−0.716	0.972 *	−0.515	0.972 *	0.073	0.999 **	0.992 **	0.997 **	0.991 **	0.992 **	1.000				
β-Car	0.981 **	0.697	0.971 **	−0.231	0.187	0.980 **	−0.432	0.981 **	0.757	0.998 **	0.994 **	0.993 **	−0.161	0.982 *	0.997 **	1.000			
(6R)-δ-Car	0.958 *	−0.546	0.979 *	−0.962 *	−0.754	0.977 *	−0.543	0.936	0.042	0.993 **	0.990 *	0.992 **	0.972 *	0.987 *	0.992 **	0.981 *	1.000		
Total Car	0.980 **	0.736	0.983 **	−0.181	0.223	0.988 **	−0.435	0.987 **	0.788	1.000 **	0.998 **	0.998 **	−0.220	0.989 *	1.000 **	0.998 **	0.989 *	1.000	
Folate	0.991 **	0.692	0.983 **	−0.224	0.165	0.985 **	−0.432	0.948 *	0.752	0.990 **	0.991 **	0.986 **	−0.176	0.955 *	0.970 *	0.982 **	0.990 *	0.985 **	1.000

** means significant correlation at 0.01 level, * means significant correlation at 0.05 level. α-T, α-T3, β-T, γ-T, γ-T3, δ-T, δ-T3 mean α-tocopherol, α-tocotrienol, β-tocopherol, γ-tocopherol, γ-tocotrienol, δ-tocopherol and δ-tocotrienol sequentially; VE means vitamin E; Lut, Zea, α-Cry, β-Cry, ε-Car, α-Car, β-Car, (6R)-δ-Car and γ-Car mean lutein, zeaxanthin, α-cryptoxanthin, β-cryptoxanthin, ε-carotene, α-carotene, β-carotene, (6R)-δ-carotene and γ-carotene sequentially.

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
