# Peer review of "Effect of Sweet Corn Residue on Micronutrient Fortification in Baked Cakes"

_foods, 2019, doi:10.3390/foods8070260_

Reviewer 1 Report

Effect of sweet corn residue on micronutrients fortification in baked cakes The manuscript reports on partly or wholly replacing wheat flour with sweet corn residue in baked cakes and the resultant effect on the micronutrients, vitamin E, carotenoids and folate. The effect of sweet corn residues on the sensory perception of baked cakes was poor but had significant impact on the levels of micronutrients. The diversity of vitamin E and carotenoids were enhanced in the sweet corn residue fortified cakes compared to the wheat flour cake. There is scope to extend the results for nutrient fortification in vulnerable groups. The manuscript has typographical and grammatical errors. The authors present Results before Materials and methods which is not consistent with journal requirement. Also the use of past and present tense needs to be corrected in the entire manuscripts. Although, it looks interesting, the data is not sufficiently justified and discussed. See specific comments: Line 21 to 23: I don’t agree with the conclusion, reason being that, replacing the flour wholly with sweet corn residue renders the cake unpalatable and hence not applicable for product formulation. Consumers must like the product to benefit from any fortification intention. Line 56: Table 1 reportsed Lines 56 and 59: Why has authors presented the same information twice, as a Table 1 and spider plot in Figure 1. The same applies to lines 67 and 68. Line 83…… Figure 3 shows represented …… Line 84: …….. which obtained from ….. Line 87: What does the area under curve imply? Authors should state this. Lines 90 - 91: “We inferred that 100 g cake could provide 279.6, 258, 194, 161.6, 119.6 and 422.4 kcal sequentially” How? Authors should justify their calculation. Line 92-93: Please rewrite and state either values were lower or a quarter of blank values. Line 97: Please provide a more suitable and informative figure caption for Figure 3. Line 103:……….in cakes are were indicated presented in Figure 4A and Table 4 Live 110: Figure 4A and B. Why did authors represent chromatograms for all samples? Choosing to represent them means authors must discuss variations within the different samples, which has not been done. I recommend removing chromatograms or just show a sample since they all look similar at most. Lines 142 – 144: Figure 7 is not discussed at all. Either discuss or delete. Lines 152-155: Authors must provide an explanation to this. Line 161: Why speculate? Please revise this statement. Line 162: This is not clear, please revise. Lines 167-170: This explanation is not appropriate. The digestion was done in vitro, but the explanation provided is in vivo which has is not supported by results presented. Lines 172 -173: State seems incomplete. Please revise. Lines 174-175: How is this achieved? Lines 197-99: How is this possible if product is not acceptable to consumers? Lines 213: Should come after the introduction. Line 5: The conclusion is not consistent with the data. For example, the texture of the products were not better than the all wheat flour.

Author Response

Dear reviewer:

Thank you for your patient for carefully revising our manuscript entitled ‘Effect of sweet corn residue on micronutrients fortification in baked cakes’ (ID: foods-525435). The given comments concerning our manuscript are useful for improving the manuscript. We have carefully studied the comments one by one and made correction that hope meet with approval. Corrections are marked in manuscript.

 For the comments in the report, we have made following revision:

1. The manuscript has typographical and grammatical errors. The authors present Results before Materials and methods which is not consistent with journal requirement. Also the use of past and present tense needs to be corrected in the entire manuscripts.

Response: We have reviewed the manuscript and revised some inappropriate portions as required by the template.

 2. Although, it looks interesting, the data is not sufficiently justified and discussed.

Response: We have discussed the experimental results further.

 3. Line 21 to 23: I don’t agree with the conclusion, reason being that, replacing the flour wholly with sweet corn residue renders the cake unpalatable and hence not applicable for product formulation. Consumers must like the product to benefit from any fortification intention.

Response: We have synthesized the results of sensory evaluation, texture analysis and nutrient content to obtain an optimal formula.

 4. Line 56: Table 1 reportsed Lines 56 and 59: Why has authors presented the same information twice, as a Table 1 and spider plot in Figure 1. The same applies to lines 67 and 68.

Response: We have deleted the repetitive spider plot.

 5. Line 83…… Figure 3 shows represented …… Line 84: …….. which obtained from ….. Line 87: What does the area under curve imply? Authors should state this.

Response: We have explained some of them in more detail.

 6. Lines 90 - 91: “We inferred that 100 g cake could provide 279.6, 258, 194, 161.6, 119.6 and 422.4 kcal sequentially” How? Authors should justify their calculation.

Response: We have supplemented the calculation basis for this part of the results.

 7. Line 92-93: Please rewrite and state either values were lower or a quarter of blank values.

Response: We have innovatively expressed this part of the quantitative relationship.

 8. Line 97: Please provide a more suitable and informative figure caption for Figure 3.

Response: We have replaced the original title with ‘Amount of reducing sugars released during in vitro digestion.’.

 9. Line 103:……….in cakes are were indicated presented in Figure 4A and Table 4 Line 110: Figure 4A and B. Why did authors represent chromatograms for all samples? Choosing to represent them means authors must discuss variations within the different samples, which has not been done. I recommend removing chromatograms or just show a sample since they all look similar at most.

Response: We have deleted these two graphs.

 10. Lines 142 – 144: Figure 7 is not discussed at all. Either discuss or delete.

Response: We have replaced the original graphs with tables and discussed it.

 11. Lines 152-155: Authors must provide an explanation to this.

Response: We have re-discussed this part in more depth.

 12. Line 161: Why speculate? Please revise this statement.

Response: Dietary fiber will not be decomposed at baking temperature. We have recounted the basis here and moved it to the conclusion that we think is more appropriate.

 13. Line 162: This is not clear, please revise.

Response: We have redefined this part of the discussion.

 14. Lines 167-170: This explanation is not appropriate. The digestion was done in vitro, but the explanation provided is in vivo which has is not supported by results presented.

Response: We have reorganized the language and further discussed the results of this part of digestion in vitro.

 15. Lines 172 -173: State seems incomplete. Please revise.

Response: We have made a supplementary explanation to this part.

 16. Lines 174-175: How is this achieved?

Response: It reduced GI by replacing sugar in the cake. We have made a clearer statement.

17. Lines 197-99: How is this possible if product is not acceptable to consumers?

Response: We have made a comprehensive discussion and evaluation of the samples and found that the best formulation can also meet the supplementary requirements.

 18. Lines 213: Should come after the introduction.

Response: We have moved this part to the right place.

 19. Line 5: The conclusion is not consistent with the data. For example, the texture of the products were not better than the all wheat flour.

Response: The sensory, texture and micronutrient contents of the samples were compared comprehensively, and the optimum formula scrc2 was obtained.

Reviewer 2 Report

I read the manuscript with great interest, however I have some suggestion how to improve it.

 Abstract

Short information concerning methods should be added to the abstract

Introduction

Line 44 and Line 45-46 – please add the reference which confirm content of folate and carotenoids in SCR. References  used in manuscript do rather confirm the influence on nutrients on human body.

Line 52 -53 – the aim should be changed. “is to increase the added value of SCR”

I suggest to complete this part of the manuscript with positive and negative consequences of modifying the composition of the product, also from the perspective of the consumer.

Materials and Methods

This chapter should be placed after Introduction and before Results.

In the current manuscript layout, there is a difficulty in understanding some of the results, e.g. how the sensory characteristics of the cakes were assessed

Line 234 Who evaluated the cake sample? How the samples were given to the members of evaluation group.

Lines 284-287 – This part needs to be completed. I expect more details.

Results

Line 61 – statement  “more intense corn flavor’ needs more attention (SCR3 and SCR4)

Line 62 – the results for overall acceptance were not presented in the manuscript.

Line 65 - under the table should be an explanation of the scale used to assess each attribute.

Line 67 – statement ‘test results’ is not understandable.

Line 78 – this information should be moved to Materials and Methods.

Line 140 - the title of Figure 6 should be changed – ‘content of folate’ instead of ‘changes of folate content’  the correlation coefficients in traditional way. Moreover, repeating the number for the same two-sided correlations is unnecessary.

Line 142 I suggest to present these data in more traditional way, i.e. in the tables without repeating the same  correlation coefficients (the second part of Figure)

Conclusions

I expected  conclusion which proportion of SCR can be recommended as the “best one”.

Author Response

Dear reviewer:

We sincerely appreciate you for your warm work concerning our manuscript entitled ‘Effect of sweet corn residue on micronutrients fortification in baked cakes’ (ID: foods-525435). Thank you for your detailed and kind assessment on the manuscript. Those suggestions are pertinent and helpful for us to improve the manuscript as well as providing significant guidance to our researches. We have carefully studied the comments and made correction and improvement in manuscript that hope will meet with approval. Corrections are marked in manuscript.

 For some aspects mentioned in report, we have revised and hope they will meet with approval.

1. Short information concerning methods should be added to the abstract

Response: We have supplemented the brief experimental methods in the abstract.

 2. Line 44 and Line 45-46 – please add the reference which confirm content of folate and carotenoids in SCR. References used in manuscript do rather confirm the influence on nutrients on human body.

Response: At present, there are no literatures directly studying carotenoids and folate in SCR, which are generally described as part of sweet corn. For this reason, we have changed our expression to make it more reasonable.

 3. Line 52 -53 – the aim should be changed. “is to increase the added value of SCR”

I suggest to complete this part of the manuscript with positive and negative consequences of modifying the composition of the product, also from the perspective of the consumer.

Response: In this section, we have made a clearer statement about the purpose of the study.

 4. This chapter should be placed after Introduction and before Results.

Response: We have placed it after Introduction and before Results.

 5. In the current manuscript layout, there is a difficulty in understanding some of the results, e.g. how the sensory characteristics of the cakes were assessed

Response: The establishment of evaluation indicators is generally based on relevant standards or mature research models, which we have also elaborated.

 6. Line 234 Who evaluated the cake sample? How the samples were given to the members of evaluation group.

Response: We have redefined the process of sensory evaluation and defined the evaluator and the way of evaluation.

 7. Lines 284-287 – This part needs to be completed. I expect more details.

Response: We have added more details about data processing.

 8. Line 61 – statement “more intense corn flavor’ needs more attention (SCR3 and SCR4)

Response: We reinterpreted the sensory experiment results in a more reasonable way.

 9. Line 62 – the results for overall acceptance were not presented in the manuscript.

Response: We have added the total score of sensory evaluation of each sample to the table presenting the experimental results.

 10. Line 65 - under the table should be an explanation of the scale used to assess each attribute.

Response: We have added the proportions of each indicator below the table.

 11. Line 67 – statement ‘test results’ is not understandable.

Response: ‘test results’ meant ‘test result with Texture Analyser’.

 12. Line 78 – this information should be moved to Materials and Methods.

Response: Material and Methods had a similar saying, and we have moved it to a more appropriate part, Results.

 13. Line 140 - the title of Figure 6 should be changed – ‘content of folate’ instead of ‘changes of folate content’

Response: The title of Figure 2 haved been changed – ‘content of folate’ instead of ‘changes of folate content’.

 14. the correlation coefficients in traditional way. Moreover, repeating the number for the same two-sided correlations is unnecessary. Line 142 I suggest to present these data in more traditional way, i.e. in the tables without repeating the same correlation coefficients (the second part of Figure)

Response: We have replaced hot spot figures with traditional tables and focused on their correlation.

 15. I expected conclusion which proportion of SCR can be recommended as the “best one”.

Response: The sensory, texture and micronutrient contents of the samples were compared comprehensively, and the optimum formula scrc2 was obtained.

Round  2

Reviewer 1 Report

I am happy with changes.

Reviewer 2 Report

Thank you very much for all improvemnets. This version of manuscript is acceptable for me.

Kind regards